# Human Umbilical Cord Mesenchymal-Stem-Cell-Derived Extracellular Vesicles Reduce Skin Inflammation In Vitro

**DOI:** 10.3390/ijms242317109

**Published:** 2023-12-04

**Authors:** Tzou-Yien Lin, Tsong-Min Chang, Wei-Cheng Tsai, Yi-Ju Hsieh, Li-Ting Wang, Huey-Chun Huang

**Affiliations:** 1Department of Paediatrics, Chang Gung Memorial Hospital, Chang Gung University College of Medicine, Taoyuan 33305, Taiwan; alinpid@gmail.com; 2Department of Hair Styling and Design, Department of Applied Cosmetology, Hungkuang University, Taichung 433304, Taiwan; ctm@sunrise.hk.edu.tw; 3ExoOne Bio Co., Ltd., Taipei City 115011, Taiwan; wayne0220@exo-one.com (W.-C.T.); vhsiehi@exo-one.com (Y.-J.H.); litingwang@exo-one.com (L.-T.W.); 4Department of Medical Laboratory Science and Biotechnology, College of Medicine, China Medical University, Taichung 404328, Taiwan

**Keywords:** extracellular vesicle, umbilical-cord-derived mesenchymal stem cells, SIRT1, p53, NF-κB

## Abstract

The protective roles of extracellular vesicles derived from human umbilical cord mesenchymal stem cells against oxazolone-induced damage in the immortalized human keratinocyte cell line HaCaT were investigated. The cells were pretreated with or without UCMSC-derived extracellular vesicles 24 h before oxazolone exposure. The pretreated UVMSC-EVs showed protective activity, elevating cell viability, reducing intracellular ROS, and reducing the changes in the mitochondrial membrane potential compared to the cells with a direct oxazolone treatment alone. The UCMSC-EVs exhibited anti-inflammatory activity via reducing the inflammatory cytokines IL-1β and TNF-α. A mechanism study showed that the UCMSC-EVs increased the protein expression levels of SIRT1 and P53 and reduced P65 protein expression. It was concluded that UVMSC-EVs can induce the antioxidant defense systems of HaCaT cells and that they may have potential as functional ingredients in anti-aging cosmetics for skin care.

## 1. Introduction

Dermatological medicine encompasses a broad spectrum of skin disorders, including psoriasis, eczema, atopic dermatitis, lichen planus, bullous pemphigoid, systemic lupus erythematosus, and chronic wound healing. It is a multidisciplinary field that aims to restore the skin’s function in cases of trauma, aging, or disease. Recent research has shown that adipose mesenchymal-stem-cell (MSC)-derived extracellular vesicles (EVs) have beneficial effects on murine skin wound models. These EVs help to promote chronic wound healing by regulating the inflammatory response, strengthening wound-healing mechanisms, and limiting scar formation. The exact mechanism by which EVs induce wound healing is still unknown [1,2,3].

EVs are naturally non-immunogenic, since they contain cell-secreted particles that enable intracellular interactions. Due to their inherent safety, EVs make an excellent choice for innovative therapies [4,5]. Our prior work showed that EVs released from umbilical cord MSCs (UCMSC-EVs) repressed mast cell activation, reduced pro-inflammatory cytokine production, and helped manage allergic disorders [6]. Current reviews have reported that EVs mediate the functions of MSCs, such as immune regulation, tissue repair, and homeostasis recovery [7]. Accelerating the wound repair of the skin is of great clinical significance, both in difficult wound diseases and in postoperative repair [8]. Further studies are needed to uncover the potential mechanism of EV-based therapeutics for wound management.

Oxidative stress causes cellular skin inflammation and senescence through ROS-related pathways, including the silent information regulator 1 (Sirtuin 1)/forkhead box O (FOXO) (SIRT1/FOXO) signaling pathways, the nuclear factor kappa B (NF-κB) signaling pathway, and the p53-related pathway [9]. Sirtuins have been the focus of much research due to their role in controlling targets associated with aging and increasing the human lifespan [10]. SIRT1, one of the human sirtuin isoforms, is an NAD+-dependent protein deacetylase that has a vast spectrum of substrates, including the important nuclear proteins p53 and FOXO3 [11]. Recent studies have revealed that SIRT1 functions as a multifunctional transcription factor that can modulate various physiopathological processes, including epithelial regeneration [12]. The SIRT1 protein modulates immune responses by deacetylating NFκB, resulting in decreased NFκB activity [13]. The NF-κB cascade can activate and regulate inflammatory factors such as IL-6 and TNF-α. These factors can act as new activators of NF-κB, forming a positive feedback mechanism for inflammatory signals. Notably, the activation of NF-κB signaling promotes cell senescence [14], while a blockade of the NF-κB pathway prolongs the life spans of fruit flies and mice [15].

P53 is a crucial protein in the regulation of the cell cycle and gene integrity under harmful conditions. It helps to maintain the homeostasis of the skin by ensuring that damaged cells are either repaired or eliminated to maintain the integrity of the epidermis. SIRT1 deacetylates p53, inhibiting its trans-activating ability and suppressing transcription [16]. We evaluated the involvement of SIRT1, p53, and NF-κB in skin damage development and assessed the beneficial effects of UCMSC-EVs on keratinocytes. Among p53-responsive genes, the transcription factors of the forkhead box O3 transcription factors (FOXO3), a subfamily of the forkhead box O transcription factors, attracted our attention because they control the expression of genes related to antioxidant stress and anti-aging [17,18]. FOXO3 is a target gene for which p53 functions as a direct transcriptional activator upstream of the FOXO3 gene [19]. These two transcriptional factors are critical to controlling both longevity and tumor suppression. However, the mechanism by which EVs regulate cellular responses has not been studied in sufficient detail. This study aimed to elucidate the potential of UCMSC-EVs as a raw material for formulating the regeneration and repair of functional skin products in the pharmaceutical and cosmetic sectors.

## 2. Results

### 2.1. EVs from UCMSCs Exhibited Exosomal Traits

The shapes and size distributions of UCMSC-EVs were observed using TEM and NTA. The results of the particle size analysis confirmed that the main peak of particle size was within the typical size range of extracellular vesicles (30–150 nm). Furthermore, the diameter of the extracellular vesicles was approximately 103.3 ± 1.0 nm as per the NTA results (Figure 1a). TEM is particularly useful for characterizing the content of extracellular vesicles, with the advantage of being label-free. TEM revealed that the UCMSC-EVs consisted of spherical double-membrane-bound vesicles (Figure 1b) and presented the exosome-positive markers CD9, CD63, and CD81 according to an immunochemical flow cytometry analysis (Figure 1c).

### 2.2. Viability of HaCaT Cells under UCMSC-EV Treatment

We initially used an MTS assay to evaluate the cell growth effects of UCMSC-EVs on HaCaT cells. HaCaT cells were added with a total number of UCMSC-EVs ranging from 1 × 10^8^ (equivalent to 1 μg/mL of total protein) to 1 × 10^10^/mL (equivalent to 100 μg/mL of total protein) before exposure to oxazolone. No significant cytotoxicity was observed after 24 h of exposure to oxazolone compared to the blank control, suggesting that low-dose oxazolone was suitable for investigating the long-term effects of keratinocyte irritation. The HaCaT cell viability was also unaffected by each dose of UCMSC-EVs (Figure 2).

### 2.3. UCMSC-EVs Accelerated Cell Migration

To explore the role of UCMSC-EVs in cell migration activity, cultured HaCaT cells were treated for 1 h with mitomycin to inhibit cell proliferation before recording the scratch gap at 24 h. The scratch wound assay indicated that the UCMSC-EVs led to a striking increase in cell migration compared to that in the control (Figure 3). These results indicated that UCMSC-EVs might facilitate skin wound healing via the acceleration of the migration of HaCaT cells.

### 2.4. Retention of UCMSC-EVs in HaCaT Cells

We detected the retention of the UCMSC-EVs in the cells by tracking the fluorescence-labeled EVs; a green signal was observed in the cytoplasm and perinuclear region of HaCaT cells at 4 h. After 1 d, the UCMSC-EVs were localized in a significant density in the nuclear zone (Figure 4). On day 3, a punctate staining pattern was evident in the nuclear compartment of the HaCaT cells. The results suggest that the transplanted UCMSC-EVs can be taken up into cells, and could penetrate and then accumulate in the cell nucleus.

### 2.5. UCMSC-EVs Protected against Oxazolone-Induced Oxidative Stress and Inflammation

The HaCaT cells were exposed to UCMSC-EVs at a 10-fold dose difference for 24 h prior to oxazolone treatment, and the levels of ROS in the intracellular and mitochondrial compartments were measured. Using a DCFDA probe that reacted with various ROS in the cells, we detected a significant elevation in the intracellular ROS levels after the oxazolone treatment, and a concomitant increase in mitochondrial superoxide was detected using a MitoSox probe that was specific for mitochondria (Figure 5). Next, we examined whether the UCMSC-EVs displayed an antioxidative benefit for keratinocytes. The addition of the UCMSC-EVs effectively diminished the endogenous ROS levels by 30% compared to the control cells. The application of 10^8^–10^10^ particles/mL of EVs revealed a dose–response effect on the elimination of ROS. Additionally, the oxazolone treatment induced the generation of mitochondrial ROS by more than 1.7-fold, and the levels of mitochondrial ROS were decreased in the UCMSC-EV-pretreated cells in a dose-dependent manner compared with the cells treated with oxazolone alone, although the magnitude of downregulation was not as significant as that for the intracellular cytoplasmic ROS. As shown in Table 1, oxazolone resulted in dramatic increases in IL-1β and TNF-α; in contrast, the expression of pro-inflammatory cytokines induced by oxazolone was significantly diminished by the pretreatment with UCMSC-EVs.

### 2.6. Effects of UCMSC-EVs on the Expression of SIRT1, p53, and p65 in HaCaT Cells

As shown in Figure 6, we observed that oxazolone triggered the activation of pro-inflammatory factor p65 in human HaCaT cells. Meanwhile, the expression of the stress protectors SIRT1 and p53 were reduced in HaCaT cells stimulated by this damage inducer. Moreover, a Western blot analysis revealed that the SIRT1 levels were significantly increased; as well, the expression of p53 was markedly augmented in the EV-primed group in comparison to the oxazolone group (*p* < 0.05). We also noted that the decline of p65 after the UCMSC-EV treatment, which contributed to the reduction in NF-κB signaling, thus repressed inflammation. The p53 tumor-suppressor protein guides cell fate decisions via controlling corresponding networks to maintain the genomic integrity in the cellular response to stresses. Taken together, these findings indicate that UCMSC-EVs effectively blocked oxazolone-induced inflammation and the modulatory response, partly through the regulation of SIRT1, p65, and p53.

### 2.7. The Binding of p53 in the FOXO3 Promoter Region Detected by Chromatin Immunoprecipitation/PCR

Considering that p53 exerts a significant protective effect against stress via antioxidant and anti-inflammatory mechanisms, we next employed a chromatin immunoprecipitation (ChIP)/qPCR assay to examine the binding of the p53 protein to FOXO3 in vivo. Two putative p53-binding sites were predicted in the FOXO3 promoter region [20,21]. The qPCR and electrophoresis results showed that the interaction between p53 and the FOXO3 promoter was weakened in the oxazolone-stimulated HaCaT cells, which was reversed by the UCMSC-EV pretreatment (Figure 7). Concomitantly, the UCMSC-EV treatment increased the p53 protein expression, thus effectively increasing the binding function on the promoter of FOXO3 to initiate its transcription activity.

## 3. Discussion

Based on the guidelines of the Minimal Information for Studies of Extracellular Vesicles [22], we performed TEM, NTA, and EV protein signature measurements to validate the physicochemical characterization of UCMSC-EVs. The UCMSC-EVs showed a spherical morphology with an average of 113 nm in diameter, and the surface biosignatures showed the positive exosomal markers CD9, CD63, and CD81.

EVs can be internalized into cells and can activate anti-inflammatory signaling by shuttling proteins and genetic contents [23]. The specific molecular mechanisms of UCMSC-EV transfer and maintenance within the recipient keratinocytes are still unclear. However, we have demonstrated that cultured keratinocytes are capable of taking up exosomes when they are added exogenously to the medium. Our confocal-microscopy-based assay allowed us to directly visualize the internalized exosomes and their persistence within keratinocytes. These observations support EVs’ ability to be internalized and transfer content to recipient cells [24]. The internalization of EVs by the keratinocytes might account for their anti-inflammatory effect. Furthermore, it is reasonable to hypothesize that the surface receptors CD9 and CD63, which are adhesion molecules, could aid in the incorporation of EVs into cells. To further investigate this possibility, research is needed to develop strategies that can enhance EV uptake and retention. Oxazolone exacerbates the cellular injury by the ROS-induced secretion of proinflammatory cytokines, which leads to an inflammatory status through the NF-κB-signaling-pathway-stimulated expression of IL-1β and TNF-α in HaCaT cells. The current study demonstrated that internalized UCMSC-EVs blunted the ROS and pro-inflammatory cytokines in HaCaT cells. This tool may, therefore, be useful, as EVs circumvent concerns about extensive expansion, cryopreservation, complications, and maldifferentiation of live replicating MSCs.

EVs are used as a scalable MSC source for the generation of pro-regenerative factors in adequate quantities [25]. Compared to the negative control group, the EV group exhibited a remarkable increase in the number of migrated cells and the wound closure rate. This signifies the potential of EVs to enhance the migration capacity and play a crucial role in the re-establishment of an intact epithelium in the injury area. While the exact cargo carried by EVs is still a mystery, exploring their interactions with cells can unlock the key to achieving successful epithelial wound healing.

The constitutive supraphysiological concentration of intracellular ROS leads to the unspecific oxidation of proteins and altered response patterns, as well as to reversible and irreversible damage to biomolecules. Controlling specific ROS-mediated signaling pathways through selective targeting offers a perspective for a more refined oxidative distress medicine. SIRT1 plays a role in maintaining cell homeostasis by managing oxidative stress, aging, metabolism, and genome protection [26,27]. The p65 subunit of NF-κB is the direct target of SIRT1, which, through deacetylation, can control the acetylation level of NF-κB p65 to regulate the transcription level of the downstream genes, including those encoding IL-1, and TNF-α inflammatory factors. The toxic damage caused by oxazolone reduced the expression of SIRT1 so that NF-κB could not be deacetylated, resulting in the activation of the inflammatory response, whereas UCMSC-EVs resisted the oxidative damage effects by inhibiting NF-κB signaling in HaCaT cells. The EVs upregulated the expression of SIRT1, further inhibiting the activity of NF-κB, terminating the release of downstream inflammatory mediators, and protecting against oxazolone-induced cytotoxicity.

As a cardinal hub surrounded by a complex molecular weave composed of multilayered regulators and effectors, p53 is a key player in this UCMSC-EV-mediated protective response [28,29]. In the current study, UCMSC-EVs efficiently defended against harmful oxidative stress by pushing the p53 levels even higher; an accumulation of p53 can lead to DNA protection. This is consistent with the prospect that p53 ensures that inflammatory responses are properly contained. Furthermore, EV treatments coordinately regulate the mutually exclusive relationship within p53 and NF-kB, revealing that EVs provide clinical translational potential for immunomodulation in wound-healing therapies [30].

The mechanistic study revealed that the binding of p53 to FOXO3 is controlled by UCMSC-EVs, which serve to prevent a decrease in p53 expression levels and to maintain the affinity between p53 and the FOXO3 gene. p53 binding triggers FOXO3 gene expression, thereby accelerating keratinocyte migration and re-epithelialization and counteracting the adverse effects of oxidative stress [31,32]. EV-induced SIRT1 also intensifies the control of FOXO3 by deacetylating FOXO3, which plays a positive role in wound healing by inducing TGF-β expression and upregulating antioxidant genes [33]. However, the negative and positive feedback loops within FOXO3 and p53 are dependent on the cell type and the specific environmental conditions. Because p53 activity can be regulated by ROS, which, in turn, regulates FOXO3, we speculate that an EV-altered redox state may occur via the p53-FOXO3 signaling axis; however, a further in-depth validation of their role is required. In conclusion, this study provides the first evidence that UCMSC-EVs have the potential to be effective and safe interventions for wound therapeutics.

## 4. Materials and Methods

### 4.1. UCMSC Culture and UCMSC-EV Isolation

UCMSCs were cultured to 90% confluence in α-MEM containing 5% human platelet lysate (Mill Creek Life Sciences, Rochester, MN, USA) and maintained in 10-layered Cell Factory Systems (Cell Factory Systems) at 37 °C and 5% CO_2_. The culture medium was harvested and filtered through a 0.22 μm polyethersulfone membrane filter (Thermo Fisher Scientific, Waltham, MA, USA) to remove large nonexosome particles and cell debris. The EVs were isolated using a tangential flow filtration (TFF) system (Sartorius Stedim Biotech, Göttingen, Niedersachsen, Germany) with a 100 kDa molecular weight cutoff filter. All operations were performed at 4 °C. The final exosome pellets were stored at −80 °C. The protein content of the exosomes was measured using a bicinchoninic acid protein assay (BCA; Thermo Fisher Scientific, Waltham, MA, USA).

### 4.2. Nanoparticle Tracking Analysis (NTA)

The size distribution and concentration of the isolated EV suspensions were analyzed using a nanoparticle tracking analysis (NTA) (NanoSight NS300). The samples diluted in PBS were then injected into the laser chamber, yielding particle concentrations in the region of 1012–1014 particles/mL, according to the manufacturer’s recommendations. All samples were analyzed in triplicate.

### 4.3. Surface Marker Analysis by Nano-Flow Cytometry (nFCM)

The EV surface markers CD9, CD63, and CD81 were analyzed using a Flow NanoAnalyzer (NanoFCM, Xiamen, China). The EVs were mixed with fluorescence-conjugated antibodies and incubated at 37 °C. After being washed with PBS and centrifuged at 100,000× *g* at 4 °C, the samples were diluted to achieve particle counts within the optimal range of 2000–12,000/min. The flow rates and side-scattering intensities were converted to the corresponding vesicle concentrations and sizes on the NanoFCM software (NanoFCM, Profession V2.0) using calibration curves.

### 4.4. Cell Viability

HaCaT cells were seeded at a concentration of 15,000 cells/mL in 96-well plates. Cell viability was assessed using an MTT assay. Viability was calculated from the absorbance ratio at 570 nm in the EV-treated cells compared to the EV-free control.

### 4.5. Cell Migration Assay

HaCaT cells (2 × 10^5^) were cultured in migration chambers. After the removal of the barrier, the cells were supplemented with fresh culture medium. We used a 10× objective lens and bright-field phase contrast for imaging the cell migration after 24 h, and three independent experiments were performed. The migration area was quantified as the area of the initial wound area—the remaining area of the wound 24 h post-wounding using Image J software (https://imagej.net/ij/download.html) (NIH, Bethesda, Maryland, USA).

### 4.6. Exosome Uptake Assay

UCMSC-EVs were labeled with a green fluorescent dye (ExoGlow™-Protein EV Labeling Kit, System Biosciences, Palo Alto, CA, USA) and then incubated with cells at 37 °C for the metering point. After, the cells were fixed in 4% paraformaldehyde for 15 min and the nuclei were stained with DAPI (0.5 μg/mL; Invitrogen, Carlsbad, CA, USA) and rhodamine-conjugated β-actin antibodies as a counter-stain. Confocal microscopy was used to detect the fluorescence signals in the HaCaT cells.

### 4.7. Cytokine Measurement

The supernatants obtained from the HaCaT cell culture in the various treatment groups were analyzed for interleukin IL-1β and tumor necrosis factor (TNF)-α with an enzyme-linked immunosorbent assay (ELISA) kit (eBioscience, San Diego, CA, USA), according to the manufacturer’s instructions.

### 4.8. Western Blot Analysis

The cells were lysed using a lysis buffer (Tris HCl, pH of 8.0 (50 mM); NaCl (100 mM); EGTA (2 mM); NaF (10 mM); β-glycerophosphate (40 mM); Triton X-100 (0.4%); aprotinin (10 μg/mL); and phenylmethylsulfonyl fluoride (PMSF, 1 mM)). A total of 20 μg of protein extract was resolved using SDS-PAGE and transferred to nitrocellulose membranes. The membranes were incubated with primary antibodies and the primary antibody was visualized using HRP-conjugated anti-mouse or anti-rabbit secondary antibodies and enhanced chemiluminescence (ECL, Amersham, UK).

### 4.9. Chromatin Immunoprecipitation (ChIP)

ChIP assays were performed to evaluate the binding of p53 to FOXO3 promoters in vivo. HaCaT cells were seeded in 10 cm plates at a density of 4 × 10^6^ cells/mL overnight. The cells were added to EVs for 24 h and then stimulated with oxazolone (50 μM) for 24 h. The cells were cross-linked with 1% formaldehyde for 10 min and incubated with glycine (0.125 M) for 5 min. The chromatin IP assay was performed using Abcam’s ChIP Kit Magnetic Kit. Nuclear extracts were collected by centrifuging at 2000 rpm for 5 min. The cell nuclei were resuspended in RIPA buffer (1 × PBS, 1% NP-40, 0.5% Na-deoxycholate, and 0.1% SDS supplemented with Roche protease inhibitor cocktail). Chromatin was sonicated 5 to 8 times, for 30 s each time, with a Sonics VirCell 130 sonicator equipped with a stepped microtip (Newtown, CT, USA). The chromatin was flash-frozen in liquid nitrogen and an aliquot was used to verify that the sonication was effective. Antibodies to the p53 or IgG control were coupled to Dynal magnetic beads coated with rabbit secondary antibody in PBS + 5 mg/mL BSA overnight at 4 °C. The chromatin extracts were precleared on beads and then incubated with the beads coupled to the p53 antibody overnight at 4 °C. The beads were washed twice in 1 mL of low-salt ChIP buffer, three times in 1 mL of high-salt ChIP buffer, four times with 1 mL of LiCl ChIP buffer, and once or twice in 1 mL of TE buffer. The chromatin complex was eluted in 50 μL of IP elution buffer at 65 °C overnight. The DNA was purified using phenol, chloroform, and PCR purification columns. Quantitative PCR was carried out in triplicate using SYBR Green (Bio-Rad or SA-Biosciences) on a 7900HT Fast Real-Time PCR machine (Applied Biosystems). The PCR reaction was performed on 5 μL of eluted DNA using a set of primers that amplified 100 bp of the FOXO3 gene:

Forward primer: TCTACGAGTGGATGGTGCGTTG

Reverse primer: CTCTTGCCAGTTCCCTCATTCTG

### 4.10. Statistical Analysis

The data were obtained from three independent experiments and are presented as the means ± standard error (SD). The results were statistically evaluated with a one-way analysis of variance (ANOVA) using the Social Sciences Statistical Package (SPSS, software, version 21). The *p*-value of the data was compared to the control and calculated using Student’s *t*-test (*p*-values of * *p* < 0.05, ** *p* < 0.01, *** *p* < 0.005).

## 5. Conclusions

We isolated EVs from UCMSCs and observed the uptake of human UCMSC-EVs by HaCaT cells. These results show the high potential of UCMSC-EVs in eliminating the inflammatory status induced by oxazolone in keratinocytes via upregulated SIRT1 expression, enhancing the P53 expression, downregulating the NF-κB protein level, and abolishing the ROS accumulation. The enhancement of the binding of P53 onto the FOXO3 promoter was noted to underlie UCMSC-EV modulation.

## Figures and Tables

**Figure 1 ijms-24-17109-f001:**
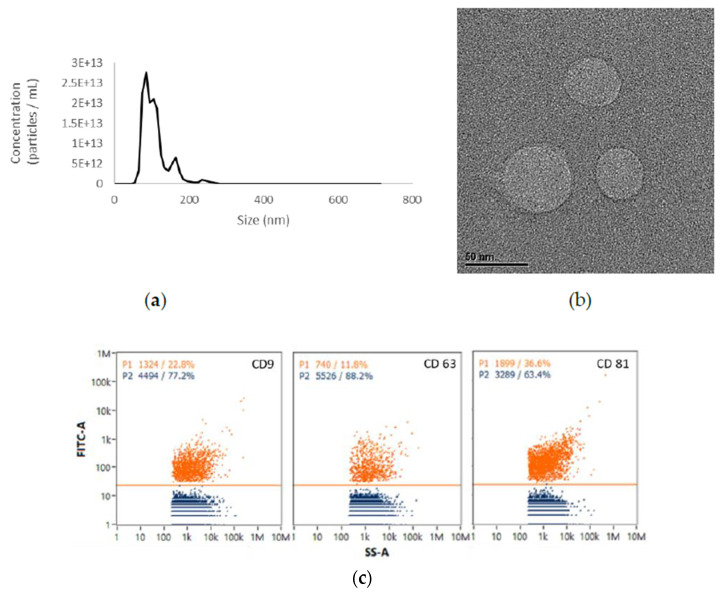
Characterization of UCMSC-EVs. (**a**) NTA analysis demonstrating the diameter of exosomes, which ranged from 55 to 145 nm, with a mean diameter of 108.3 ± 1.9 nm. (**b**) Representative images showing the morphology of UCMSC-EVs using transmission electron microscopy. Scale bar = 50 nm. (**c**) Expression of exosome markers (CD9, CD63, and CD81) examined using flow cytometry.

**Figure 2 ijms-24-17109-f002:**
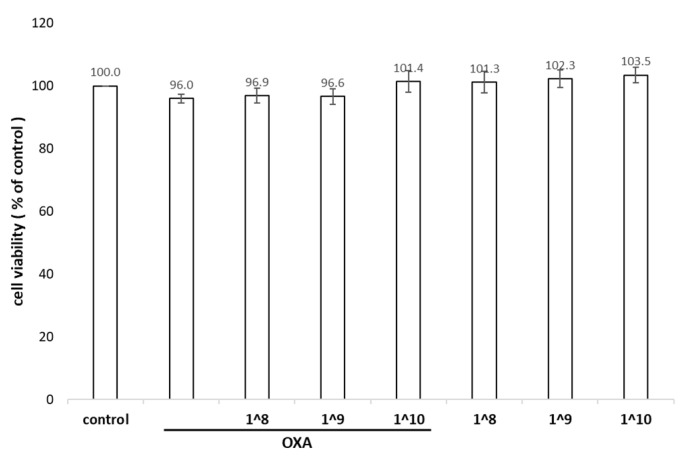
Effect of different doses of UCMSC-EVs on cell viability. HaCaT cells were incubated with different concentrations of EVs for 24 h followed by 24 h of oxazolone treatment. The results show the percentage relative to the control. Mean ± SD of three independent experiments.

**Figure 3 ijms-24-17109-f003:**
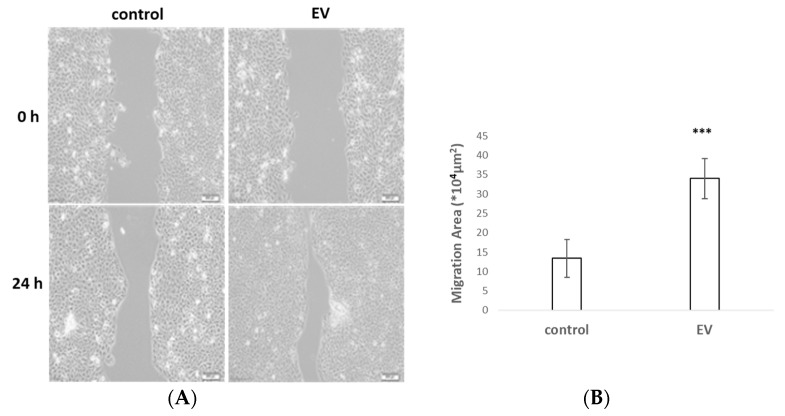
UCMSC-EVs promoted the migration of HaCaT cells. (**A**) Scratch wound assay for cells treated with EVs at 2 time points. (**B**) Statistical analysis of migration area. *** *p* < 0.001.

**Figure 4 ijms-24-17109-f004:**
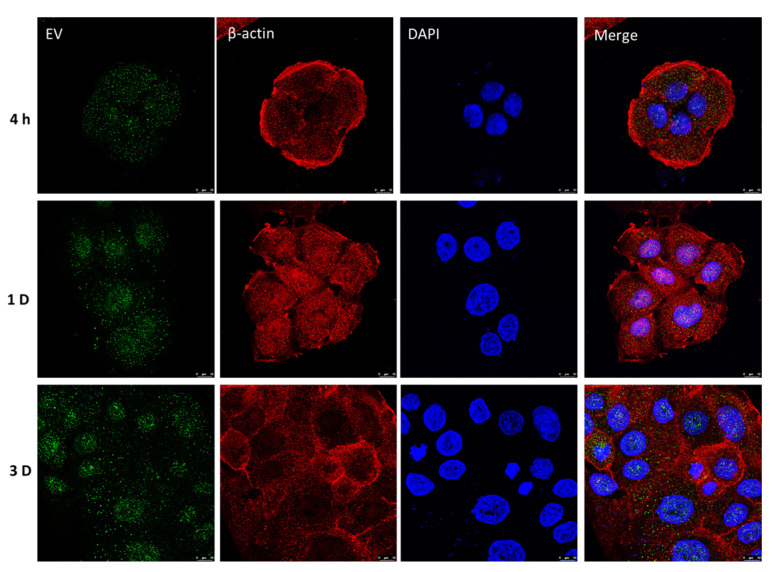
Confocal microscopy analysis of EV internalization by HaCaT cells at different time points. The green-labeled EVs were visible in the perinuclear region of cells versus the β-actin cytoskeleton (red), while blue indicates nuclei. Scale bar, 10 μm.

**Figure 5 ijms-24-17109-f005:**
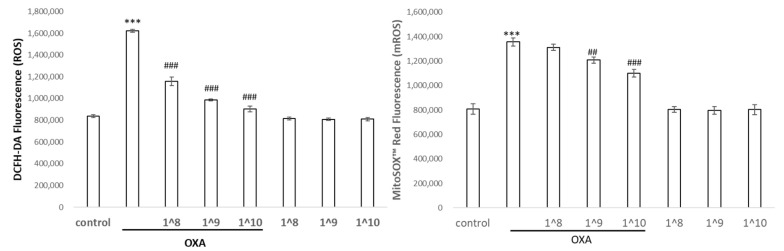
HaCaT cells exposed to oxazolone were pretreated with UCMSC-EVs at 3 doses (10^8^, 10^9^, and 10^10^) and ROS levels were measured at the indicated time points by incubating with H2DCFDA or MitoSox fluorescent probes. Data represent the mean ± SD of three independent experiments. *** *p* < 0.001 compared with the untreated control, and ^##^ *p* < 0.01 and ^###^ *p* < 0.001 com-pared with oxazolone.

**Figure 6 ijms-24-17109-f006:**
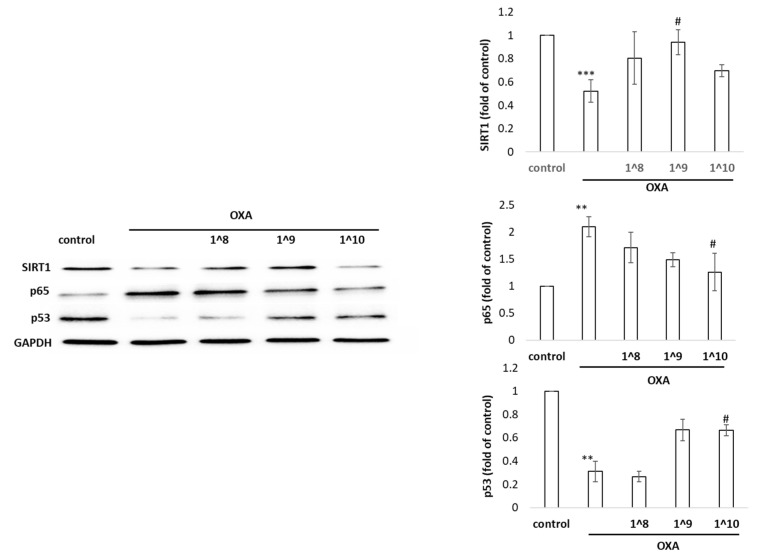
Western blot analysis of the protein expression of p53, p65, and SIRT1 in HaCaT cells administered with UCMSC-EVs. Data are shown as mean ± SD. ** *p* < 0.05 and *** *p* < 0.001 compared with the untreated control; ^#^ *p* < 0.01 compared with oxazolone.

**Figure 7 ijms-24-17109-f007:**
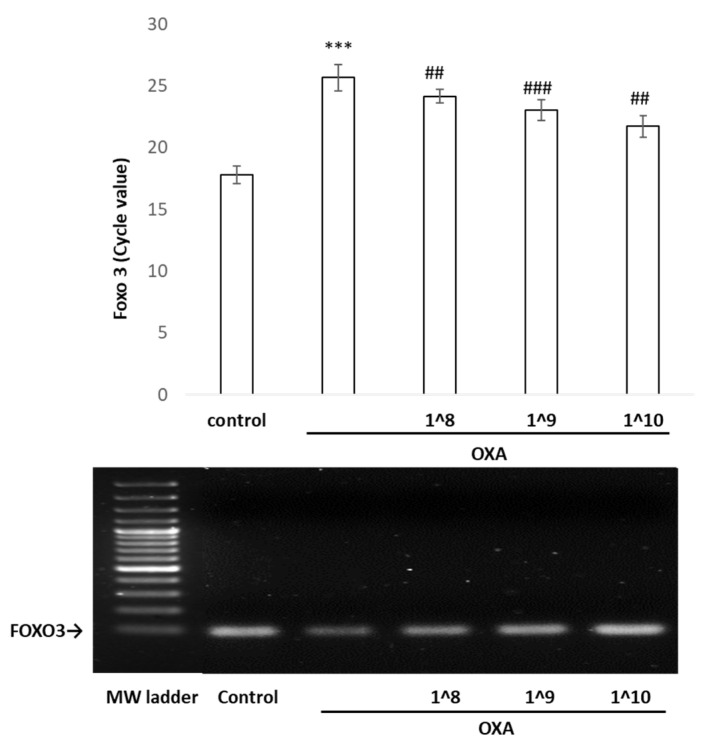
ChIP-qPCR analysis to analyze the binding ability of p53 to FOXO3 promoter. Upper panel was qPCR performed using the FOXO3 forward and reverse primer set. Data are represented as mean ± standard deviation for three independent replicates. In the lower panel, there is a representation of the PCR amplification of the FOXO3 promoter that was bound by p53. This provides more detailed insight into the increasing binding affinities of p53 to the FOXO3 promoter with higher concentrations of EV treatment. *** *p* < 0.001 compared with the untreated control, and ^##^ *p* < 0.01 and ^###^ *p* < 0.001 compared with oxazolone.

**Table 1 ijms-24-17109-t001:** Cytokine levels in oxazolone-treated and UCMSC-EV+oxazolone-treated HaCaT cells.

	Control	OXA	OXA + 10^8^	OXA + 10^9^	OXA + 10^10^
Il-1β pg/mL	18.2 ± 0.02	180.3 ±0.03 ***	129.4 ± 0.01 ^###^	111.2 ± 0 ^###^	102.2 ± 0.02 ^###^
TNF-α pg/mL	39.6 ± 0.01	192.2 ± 0.01 ***	134.4 ± 0.03 ^##^	122.8 ± 0.01 ^###^	107.5 ± 0.01 ^###^

Compared to control group, *** *p* < 0.001. Compared to oxazolone-only group, ^##^ *p* < 0.01 and ^###^ *p* < 0.001.

## Data Availability

Data are contained within the article.

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
