# Peer review of "Human Umbilical Cord Mesenchymal-Stem-Cell-Derived Extracellular Vesicles Reduce Skin Inflammation In Vitro"

_ijms, 2023, doi:10.3390/ijms242317109_

Round 1

Reviewer 1 Report

Comments and Suggestions for Authors

In the manuscript ijms-2722793 “Human umbilical cord mesenchymal stem cell-derived extracellular vesicles reduce skin inflammation in vitro”, Lin et al demonstrated that the high potential of UCMSC-EVs on anti-inflammatory activity was regulated by p53-FoxO3 signalling axis. However, some minor revisions are needed before it can be published.

Major comments:

1. A simple schematic diagram to illustrate how the pathway is regulated would be better. Since at the end, it is a bit confusing which signalling pathway(s) the author referring to.

2. The author mentioned that p53-FoxO3 signalling axis is being regulated. What is the significance of this finding? How to further improve the efficacy of the EVs based on the finding?

3. The culture media contains hPL, will it affect the overall purity and efficacy of the EVs?

Comments:

Introduction:

In general, it was easy to read, and I felt the flow was good.

In line 57-58 and 66-68: its seems that line 57-58 "we herein evaluated..." should put before the sentence in line 66 "This study aimed..." to make the aim and objective of the study clearer presented.

Results:

1. In general, it is well presented. However, all the bar chat are not clear, the x/y-axis and the lines are very pale in colour. The quality of all figures should be improved. 

Discussion:

1. Please discuss/speculate how to enhance the EVs efficacy based on the findings.

2. In line 190, is 10% considered good?

Materials and Methods:

1. Line 251: why used 1% O2?

2. Line 278; 2x105? should be 2x105

3. Howe to preserve the EVs pellet in solution form after redissolved in PBS for further testing?

Author Response

Dear reviewers of IJMS

Thank you for reviewing our manuscript entitled “Human umbilical cord mesenchymal stem cell-derived extracellular vesicles reduce skin inflammation in vitro”. Our revisions in response to the reviewers’ comments and concerns are addressed below in a point-by-point manner accordingly. We appreciate the time and effort that the reviewers have dedicated to providing valuable feedback on our manuscript. We are looking forward to your positive decision on our article.

Reviewer 2 Report

Comments and Suggestions for Authors

The Ms by Lin et al. investigated the effect of EVs recovered from  human umbilical cord mesenchymal stem cells in preventing human keratinocyte cell lines HaCaT cells from oxidative damage. They investigated the contribute of SIRT1, p53 and NFkB (p65) in their mechanism of actions.

Concerns

Material and Methods should be allocated before or after the conclusion. The conclusion must be positioned in the discussion section.

The legend to Fig 5 is not clear. The same for the figure.

The second line of Fig 6a correspond to what?

The ability of SIRT1 to act on the acetylation of p53 must be demonstrate using a specific antibody.

SIRT1 blockade must be performed to validate the results.

Comments on the Quality of English Language

The language is ok 

Author Response

(The authors gave the same response as above.)

Reviewer 3 Report

Comments and Suggestions for Authors

The introduction section is a little bit too short and it is very focused on the ideas addressed by the experimental part. I would suggest extending the description with the role of MSC exosomes and their contribution to regeneration, more examples related to UCMSCs derived exosomes, as well as a more convincing presentation of the aim of the study. The authors mention just one phrase at the end of the introduction related to the aim of the study which lacks accents on the main features or hypothesis of their study.

Line 90- please use superscript “1 *108 (equivalent to 1 μg/mL total protein) to 1 * 1010 / ml (equivalent to 100 μg/mL total 90 protein)”

The scale bar in fig 4 is not visible, please increase the font size.

Legends of figures 4 and 5 do not correspond with the figures, probably they are reversed.

Figure 7 is not clear at all, there is no indication about the PCR product dimension or any other feature. The legend says „Lower panel display immunoprecipi- 178 tated DNA amplified by PCR to compare the FOXO3 promoter fragment enrichment with or with- 179 out EV treatment”. Which is which? Very low quality of data presentation to the readers!

The protein expression data in figure 6 (western blot) should be also accompanied by genee xpression data by qPCR for the same markers.

The ChIP data is very poorly presented and explored.

The conclusions are too poorly presented and much too ambitious related to the results presented. I suggest the authors should reconsider their conclusions.

Also there are a lot of spelling mistakes and generally the English level should be improved throughout the manuscript:

„These results show the high potential of UCMSC-EVs in eliminate inflammatory status”

 „The enhancement of binding of P53 onto FOXO3 promoter were noted underlie

Comments on the Quality of English Language

there are a lot of spelling mistakes and generally the English level should be improved throughout the manuscript

Author Response

(The authors gave the same response as above.)

Reviewer 4 Report

Comments and Suggestions for Authors

The authors studied the effect of extracellular vesicles against oxazolone induced damage of keratinocytes.

The paper is well organized, the number of methods is sufficient.

I have several comments and questions:

Why as a source of damage was oxazolone? Does it have a negative effect in cosmetics or in drugs?

The quality of figures in low.

Line 90 - 8 and 10 should be written in subscript.

What do 1^8, 1^9, 1^10 in Figures 2 and 5 mean? Which substance was evaluated?

Figure 3 - cells are not seen.

Line 205 - What was a negative control?

Conclusion - not our study successfully isolated... 

but you successfully isolated

Author Response

(The authors gave the same response as above.)

Round 2

Reviewer 2 Report

Comments and Suggestions for Authors

Even improved the MS still report very preliminary data.

Comments on the Quality of English Language

The English has been improved

Reviewer 3 Report

Comments and Suggestions for Authors

The Authors have generally answered all comments and suggestions, therefore the paper appears to be acceptable for publication.

Comments on the Quality of English Language

Minor English corrections are needed.